# Apolipoprotein-CIII O-Glycosylation Is Associated with Micro- and Macrovascular Complications of Type 2 Diabetes

**DOI:** 10.3390/ijms25105365

**Published:** 2024-05-14

**Authors:** Annemieke Naber, Daniel Demus, Roderick C. Slieker, Simone Nicolardi, Joline W. J. Beulens, Petra J. M. Elders, Aloysius G. Lieverse, Eric J. G. Sijbrands, Leen M. ‘t Hart, Manfred Wuhrer, Mandy van Hoek

**Affiliations:** 1Department of Internal Medicine, Erasmus MC University Medical Center Rotterdam, P.O. Box 2040, 3000 CA Rotterdam, The Netherlands; a.naber@erasmusmc.nl (A.N.);; 2Center for Proteomics and Metabolomics, Leiden University Medical Center, P.O. Box 9600, 2300 RC Leiden, The Netherlandsm.wuhrer@lumc.nl (M.W.); 3Department of Cell and Chemical Biology, Leiden University Medical Center, P.O. Box 9600, 2300 RC Leiden, The Netherlands; 4Department of Epidemiology and Data Science, Amsterdam UMC, Location Vrije Universiteit Amsterdam, P.O. Box 7057, 1007 MB Amsterdam, The Netherlands; 5Amsterdam Public Health, Amsterdam Cardiovascular Sciences, Meibergdreef 9, 1105 AZ Amsterdam, The Netherlands; 6Department of General Practice, Amsterdam Public Health Institute, Amsterdam UMC, Location VUmc, P.O. Box 7057, 1007 MB Amsterdam, The Netherlands; 7Department of Internal Medicine, Maxima Medical Center, P.O. Box 90052, 5600 PD Eindhoven, The Netherlands; 8Department of Biomedical Data Science, Section Molecular Epidemiology, Leiden University Medical Center, Postal Zone S5-P, P.O. Box 9600, 2300 RC Leiden, The Netherlands

**Keywords:** apolipoprotein C-III, glycomics, polypeptide N-acetylgalactosaminyltransferase, diabetes complications, diabetic retinopathy, diabetic neuropathies, cardiovascular diseases, diabetes mellitus type 2

## Abstract

Apolipoprotein-CIII (apo-CIII) inhibits the clearance of triglycerides from circulation and is associated with an increased risk of diabetes complications. It exists in four main proteoforms: O-glycosylated variants containing either zero, one, or two sialic acids and a non-glycosylated variant. O-glycosylation may affect the metabolic functions of apo-CIII. We investigated the associations of apo-CIII glycosylation in blood plasma, measured by mass spectrometry of the intact protein, and genetic variants with micro- and macrovascular complications (retinopathy, nephropathy, neuropathy, cardiovascular disease) of type 2 diabetes in a DiaGene study (*n* = 1571) and the Hoorn DCS cohort (*n* = 5409). Mono-sialylated apolipoprotein-CIII (apo-CIII_1_) was associated with a reduced risk of retinopathy (β = −7.215, 95% CI −11.137 to −3.294) whereas disialylated apolipoprotein-CIII (apo-CIII_2_) was associated with an increased risk (β = 5.309, 95% CI 2.279 to 8.339). A variant of the *GALNT2*-gene (rs4846913), previously linked to lower apo-CIII_0a_, was associated with a decreased prevalence of retinopathy (OR = 0.739, 95% CI 0.575 to 0.951). Higher apo-CIII_1_ levels were associated with neuropathy (β = 7.706, 95% CI 2.317 to 13.095) and lower apo-CIII_0a_ with macrovascular complications (β = −9.195, 95% CI −15.847 to −2.543). In conclusion, apo-CIII glycosylation was associated with the prevalence of micro- and macrovascular complications of diabetes. Moreover, a variant in the *GALNT2*-gene was associated with apo-CIII glycosylation and retinopathy, suggesting a causal effect. The findings facilitate a molecular understanding of the pathophysiology of diabetes complications and warrant consideration of apo-CIII glycosylation as a potential target in the prevention of diabetes complications.

## 1. Introduction

Type 2 diabetes and its micro- and macrovascular complications pose worldwide problems in terms of morbidity, mortality, healthcare costs, and low quality of life [1]. Despite treatment efforts, a substantial residual risk of these complications remains [2,3]. Diabetic dyslipidaemia is one of the main risk factors for complications in type 2 diabetes and is characterised by a severely atherogenic lipid profile [4]. Apolipoprotein-CIII (apo-CIII) levels have been linked to dyslipidaemia [5].

Apo-CIII is a protein involved in the metabolism of triglyceride-rich lipoproteins (TRLs) [6]. It has detrimental effects on the vascular wall by reducing the clearance of TRLs via lipoprotein lipase (LPL)-dependent and -independent pathways [7]. Furthermore, apo-CIII enhances monocyte adhesion to the endothelium [8] and the binding of apoB-containing lipoproteins to vascular proteoglycans [9]. High levels of apo-CIII are found in individuals with diabetes mellitus compared with individuals without diabetes [10,11] and are related to reduced insulin sensitivity [12] and apoptosis of pancreatic beta-cells [13]. High apo-CIII levels have also been associated with diabetic retinopathy and nephropathy [14,15,16] and with increased cardiovascular disease risk in the general population and type 2 diabetes [17]. Genetic studies support the causality of this relationship with ischemic cardiovascular disease [18]. Consequently, it has been suggested that apo-CIII levels could be reduced by anti-apo-CIII antibody, antisense RNA, and silencing RNA therapies to reduce vascular disease risk [19,20,21,22].

Posttranslational modifications, such as enzymatic glycosylation, can alter the function of apolipoproteins [23,24,25]. Apo-CIII exists in four main proteoforms: the native, non-glycosylated proteoform, and *O*-glycosylated proteoforms with either zero, one, or two sialic acids [12,26]. It has been recognised that glycosylation influences apo-CIII function and its relationship with lipid metabolism and cardiovascular disease [12,25,27]. Non- and monosialylated apo-CIII are associated with the formation of small-dense low-density lipoproteins (LDL) [28]. Altered apo-CIII glycosylation affects the inhibition of LPL by apo-CIII [24] and the interaction of LDL with the vascular wall [23]. Moreover, apo-CIII glycoforms are differentially cleared by hepatic receptors [25]. Different glycosylation patterns of apo-CIII have been observed in liver disease, metabolic syndrome, and the associated factors of body weight and insulin sensitivity [29,30]. Aberrant apo-CIII glycosylation patterns might increase the risk of complications in type 2 diabetes. These glycosylation patterns may have implications for the efficacy of preventive therapies addressing diabetes complications and in particular for lipid-lowering agents in type 2 diabetes. To our knowledge, apo-CIII glycosylation, its genetic background, and its associations with complications of diabetes have not been studied before.

In the present study, we investigated the relation of apo-CIII glycosylation with the prevalence and incidence of diabetic retinopathy, nephropathy, neuropathy, and macrovascular complications. The direction of these relationships was assessed with recently identified genetic variants associated with apo-CIII glycosylation. We used data from the DiaGene study, a prospective study of type 2 diabetes conducted in Eindhoven, the Netherlands [31]. Genetic associations were replicated and meta-analysed in an independent cohort with type 2 diabetes patients in West-Friesland, the Netherlands: the Hoorn Diabetes Care System (Hoorn DCS) [32].

## 2. Results

### 2.1. Cohort Characteristics

The characteristics of cases with type 2 diabetes of both cohorts are presented in Table 1; 45.8% and 44.6% were female, and the mean age was 65.1 (SD 10.6) and 61.1 (SD 11) years in the DiaGene and Hoorn DCS, respectively. Compared with the Hoorn DCS, individuals from the DiaGene study had a longer duration of type 2 diabetes (8.0 vs. 0.6 years). At inclusion, the prevalence of complications was higher in the DiaGene than in the Hoorn DCS for retinopathy (17.4 vs. 3.6%), nephropathy (22.1 vs. 9.2%), and macrovascular complications (40.4 vs. 2.1%). In the DiaGene study, 30.6% of the participants had neuropathy at inclusion, but the Hoorn DCS did not collect data on neuropathy. The incidence of complications during follow-up was comparable between the DiaGene and the Hoorn DCS (Table 1).

### 2.2. Associations of Apo-CIII Glycosylation with Micro- and Macrovascular Complications in the DiaGene Study

Apo-CIII_1_ and apo-CIII_2_ levels showed strong associations with the prevalence (Table 2) and incidence of retinopathy (Table 3). In Model 1, apo-CIII_1_ was associated with a decreased prevalence of retinopathy (β = −7.215, 95% CI −11.137 to −3.294), while apo-CIII_2_ was associated with an increased prevalence (β = 5.309, 95% CI 2.279 to 8.339). These associations lost significance in Model 2, and the effect size was slightly reduced but showed the same trend. The association of apo-CIII_0a_ with an increased prevalence of retinopathy reached suggestive significance in Model 2 (β = 9.968, 95% CI 1.437 to 18.499), after adjustment for duration of diabetes and HbA1c. Apo-CIII_2_ was significantly associated with incident retinopathy in Model 1 (β = 4.484, 95% CI 1.158 to 7.810). The associations of apo-CIII_1_ and apo-CIII_2_ with prevalent and incident retinopathy had consistent directions of effect.

For nephropathy, apo-CIII_1_ was associated with a decreased prevalence (β = −3.945, 95% CI 7.716 to −0.174), but did not reach Bonferroni-corrected significance (*p* = 0.040) (Table 2). Adding covariates in Model 2 did not alter the direction of the association. Also, the direction and magnitude of the effect for the association of apo-CIII_1_ and incident nephropathy were similar to the association found at baseline (Table 3).

Apo-CIII_1_ and apo-CIII_2_ levels were associated with neuropathy (Table 2). Apo-CIII_1_ was significantly associated with an increased prevalence of neuropathy (β = 7.706, 95% CI 2.317 to 13.095), while apo-CIII_2_ was associated with a decreased prevalence of neuropathy without reaching Bonferroni-corrected significance (β = −4.968, 95% CI −9.065 to −0.871, *p* = 0.017). The direction of effects and their significance were similar in Model 2. The analysis for incident neuropathy did not reach significance but showed a similar direction of effects (Table 3).

Apo-CIII_oa_ levels were associated with macrovascular complications of diabetes (Table 2). This glycoform had a significant negative association with the prevalence of macrovascular complications in Model 1 (β = −9.195, 95% CI −15.847 to −2.543), which remained significant in Model 2 and the sensitivity analysis for insulin use (Table 2 and Appendix A). This association lost significance in the analysis of the incidence of macrovascular complications (Table 3). Nevertheless, the effect size and trend were similar.

The use of lipid-reducing agents was associated with lower relative levels of apo-CIII_0a_ and insulin use was associated with lower apo-CIII_1_ and higher apo-CIII_2_ (Appendix A). Most associations persisted after sensitivity analysis for the use of lipid-lowering medication and insulin (Appendix A).

### 2.3. Apo-CIII Glycosylation-Associated Genetic Variants and Complications of Diabetes in the DiaGene and Hoorn DCS Studies, a Meta-Analysis

The genetic variant rs4846913-A located in the *GALNT2* gene, previously associated with decreased apo-CIII_oa_ [36], was found to be negatively associated with prevalent retinopathy in Model 2 in the DiaGene study (OR = 0.739, 95% CI 0.575 to 0.951). However, it did not reach the Bonferroni significance level (Table 4, Table 5, Figure 1). This association remained in the sensitivity analysis for the use of lipid-lowering medications (OR = 0.722, 95% CI 0.559 to 0.932) and lost its significance after the sensitivity analysis for insulin use (Appendix A). These associations had the same direction of effect in the Hoorn DCS cohort.

The variant rs3213497-T located at an exonic non-coding RNA region in the *GALNT2:RP5-956O18.3* gene, previously associated with apo-CIII_0a_ [36], was associated at a nominal significance level with a decreased incidence of retinopathy in our meta-analysis of the DiaGene and Hoorn DCS in Models 1 and 2 (HR = 0.830, *p* = 0.009 and HR = 0.834, *p* = 0.012 resp.) (Table 6 and Table 7). This association remained after the sensitivity analysis for lipid-lowering medication and insulin use, although without Bonferroni significance level (Appendix A). The same variant rs3213497-T was nominally significantly associated with decreased prevalence of nephropathy in the meta-analysis with OR = 0.806, *p* = 0.017 (Table 5).

## 3. Research Design and Methods

### 3.1. Study Design

Plasma samples from the DiaGene study were used. In short, the DiaGene study is an all case-control study addressing all lines of healthcare with prospective follow-up in the cities of Eindhoven and Veldhoven, the Netherlands. The DiaGene study sought to include all type 2 diabetes patients in hospital and primary care at a given time point in the past. A prospective follow-up was performed from that moment onwards. Inclusion took place between 2006 and 2011, with a median follow-up duration of 7.8 years. This study has been described in detail elsewhere [31]. The current study was restricted to cases with type 2 diabetes for which apo-CIII glycosylation measurements were available (n = 1571).

Data from the Hoorn DCS (n = 5409) were used for replication of genetic associations of apo-CIII proteoforms with complications of type 2 diabetes. Only participants from the Hoorn DCS study that had GWAS data available were included in our analyses. The cohort has been described in detail elsewhere [32]. In short, biobanking materials and data on annual examinations for micro- and macrovascular complications have been collected from primary care type 2 diabetes patients, with prospective follow-up. The Hoorn DCS study included newly diagnosed type 2 diabetes patients. Afterwards, follow-up was performed. Inclusion took place between 1998 and 2014 in the region of West Friesland in the Netherlands. The median follow-up duration was 9.0 years.

All participants gave their written informed consent. Both studies were approved by the Medical Ethics Committees of the involved hospitals in compliance with the Declaration of Helsinki principles (DiaGene MEC-2004-230, Hoorn DCS 2007/57).

### 3.2. Definitions

Type 2 diabetes was defined in accordance with the American Diabetes Association and the WHO guidelines [37,38] for both the DiaGene and the Hoorn DCS cohorts [31,32]. People with other types of diabetes were excluded from both cohorts.

Retinopathy, nephropathy, and neuropathy were considered microvascular complications. Diabetic retinopathy was diagnosed using fundus photography and scored as absent or present by an ophthalmologist. Diabetic nephropathy was defined as albumin/creatinine ratio (ACR) ≥ 2.5 for men or ≥3.5 for women present at two of three consecutive measurements, or when ACR ≥ 12.5 for men or ≥17.5 for women was present at one measurement. Diabetic neuropathy was defined by a podiatrist, neurologist, or the patient’s treating physician, based on physical examination including a 10 g monofilament test and vibratory sensation test using a 128 Hz tuning fork. Only hospital-treated participants of the DiaGene study had information about diabetic neuropathy (n = 611). Macrovascular complications comprised ischemic heart disease (myocardial infarction, percutaneous coronary intervention, or coronary artery bypass graft), cerebrovascular accident, transient ischemic attack, and peripheral arterial disease. In the DiaGene, information on cardiovascular disease for hospital-treated patients was retrieved from the medical records, and for primary care-treated patients from self-reporting [31]. In the Hoorn DCS, cardiovascular events were reported during the annual visits and verified against the medical records from the regional hospital and general practitioners [32]. More detailed information on the definitions and data collection of micro- and macrovascular complications of diabetes has been described previously [31,32].

### 3.3. Apo-CIII Glycosylation Analysis

Ultrahigh-resolution matrix-assisted laser desorption/ionization Fourier transform ion cyclotron resonance mass spectrometry (MALDI FT-ICR MS) method, described elsewhere [39], was applied to assess the relative abundances of apo-CIII proteoforms in blood plasma samples of the DiaGene study. Apo-CIII exists in four main proteoforms: glycosylated variants containing a mucin-type core-1 O-glycan with either zero, one, or two sialic acids (apo-CIII_0c_, apo-CIII_1_, and apo-CIII_2_, respectively) and a non-glycosylated variant (apo-CIII_0a_) (Figure 2) [12,26]. The sum of apo-CIII_0c_, apo-CIII_1_, and apo-CIII_2_ was set to 1.0 to obtain the proportion of these three glycoforms within all glycosylated species of apo-CIII. These normalised relative peak intensities of the glycosylated variants apo-CIII_0c_, apo-CIII_1_, and apo-CIII_2_ were used to assess associations with apo-CIII sialylation status. The relative peak intensity of apo-CIII_0a_, normalised to the sum of all four proteoforms (apo-CIII_0a_, apo-CIII_0c_, apo-CIII_1_, and apo-CIII_2_), was used to assess associations with apo-CIII glycosylation status. We used beta to present the outcome measures of our regression models on glycosylation traits to ease the interpretation. Quality control of mass spectrometry (MS) data was performed as described elsewhere [39], see Appendix A. Samples not meeting the acceptable quality parameters were excluded from the analysis. Apo-CIII glycosylation data were only available in the DiaGene study.

### 3.4. Experimental Design and Statistical Analysis

A total of 1571 DiaGene samples passed apo-CIII glycosylation data quality control and contained sufficient clinical information for the analysis. Missing data on covariates (duration of diabetes, HbA1c, use of lipid-lowering medication, and use of insulin) within the DiaGene study were imputed using multiple imputations by predictive mean matching in SPSS. The maximum count of imputations per variable was 105 in DiaGene; all imputed variables had <7% missing values (Appendix A). Covariates within the DCS Hoorn had <2% missing values per covariate, and therefore were not imputed (Appendix A).

Associations of apo-CIII glycosylation with micro- and macrovascular complications at baseline and follow-up were investigated using logistic regression and Cox proportional hazards models, respectively. Prospective analyses were performed after excluding prevalent complications. Analyses were performed for retinopathy, nephropathy, neuropathy, and macrovascular complications separately.

Two models were applied for all analyses. Model 1 was adjusted for age and sex. Model 2 was adjusted for age, sex, haemoglobin A1c (HbA1c), and duration of diabetes. The outcomes of Model 1 reflect the broad differences between diabetes patients with and without one of the complications. In contrast, Model 2 reflects the differences not mediated through the duration of type 2 diabetes or its regulation, with HbA1c as a proxy.

We performed two-sided *t*-tests to analyse the associations of insulin use and lipid-lowering therapy with apo-CIII glycosylation. Subsequently, we performed two sensitivity analyses: adding insulin use to Model 1, and the use of lipid-lowering therapy (fibrates and statins) to Model 2.

In our previous genome-wide association study (GWAS) on apo-CIII glycosylation [36], we identified genetic variants associated with apo-CIII O-glycosylation. We selected two loci from this GWAS, with previous links to triglyceride levels. We analysed the associations of these genetic variants with micro- and macrovascular complications, applying the same models as described above. These analyses were replicated in the Hoorn DCS cohort and meta-analysed using a random effects model for the two cohorts using the package ‘meta’ version 7.0-0 [40]. Heterogeneity between cohorts was also evaluated using I^2^ values (Appendix A) [41].

Statistical analyses within the DiaGene study were performed using SPSS version 25. Analyses of the Hoorn Diabetes Care System (DCS) cohort and the meta-analysis were performed using R version 4.0.5. For the analysis with apo-CIII glycosylation, the Bonferroni corrected *p*-value for significance was calculated as 0.013 (0.05/4) based on the four main apo-CIII proteoforms. For the analyses of the four genetic variants and complications of diabetes, we used a Bonferroni corrected *p*-value for a significance of 0.013 (0.05/4). Significance is given after Bonferroni correction unless stated otherwise.

## 4. Discussion

In the present study, we found that apo-CIII glycosylation (specifically, sialylation) and the linked *GALNT2*-gene variant were associated with the prevalence and incidence of diabetic retinopathy. This suggests that glycosylation determines apo-CIII function and decreased glycosylation of apo-C-III contributes to the development of retinopathy. Further, apo-CIII glycosylation was associated with prevalent diabetic neuropathy and macrovascular complications.

Type 2 diabetes is characterised by insulin resistance and an inadequate compensatory insulin secretory response, resulting in chronic hyperglycaemia [42]. Glucose and insulin oppositely modulate the expression of apo-CIII [43,44]. This could explain the high apo-CIII secretion rate in the presence of type 2 diabetes [45]. So far, no other studies have investigated the relationships between apo-CIII glycosylation, its genetic background, and the complications of type 2 diabetes. Here, the current study findings are discussed for each complication and potential underlying mechanisms will be hypothesised considering the limited pathophysiological information that is available in the current literature.

The most pronounced finding of this study was the connection between *GALNT2*, apo-CIII glycosylation, and diabetic retinopathy (Figure 1). Previously, we identified a genetic variant rs4846913-A, located in the *GALNT2* gene, that was negatively associated with non-glycosylated apo-CIII (apo-CIII_0a_) [36]. The A-allele of this rs4846913 variant is known to increase the expression of the *GALNT2*-gene [46], which encodes a GalNAc-transferase that initiates mucin-type O-glycosylation of peptides, such as apo-CIII [47]. The rs4846913-A variant comes with higher glycosylation levels of apo-CIII. Although only within the DiaGene study, this variant was associated with a lower risk of retinopathy. In line, higher relative levels of apo-CIII_0a_ were associated with increased prevalence of retinopathy. Replication in a larger cohort is needed to confirm whether increased activity of *GALNT2* reduces the risk of retinopathy through glycosylation of apo-CIII. This might be an interesting target for prevention or treatment of diabetic retinopathy.

Diabetes complications develop through an interplay of risk factors, including glucose and lipid metabolism, in which apo-CIII plays a role. Previously, we analysed apo-CIII glycosylation profiles in over 700 people without diabetes [39]. Our findings supported other studies demonstrating a positive association between the apo-CIII_1_/apo-CIII_2_ ratio and triglyceride levels, implying the involvement of apo-CIII sialylation in an impaired triglyceride clearance [12,25,48]. In line, the removal of sialic acids from apo-CIII by neuraminidase treatment decreases its potential to inhibit LPL [24]. A recent study by Kegulian et al. has shown that the two sialylated glycoforms (apo-CIII_1_ and apo-CIII_2_) are cleared differently by hepatic receptors, specifically heparan sulphate proteoglycans (HSPGs) such as syndecan-1 (SDC1), LDL-receptor (LDLR), and LDL receptor-related protein 1 (LRP1) [25]. In people with type 2 diabetes, the LDL particles carry more apo-CIII_2_ and less apo-CIII_1_ than in those without diabetes [23]. Molecular mechanisms behind the association of apo-CIII_1_ and apo-CIII_2_ with triglyceride levels and vascular disease have not been fully elucidated. Mauger et al. found a positive association of small dense LDL with the production rate of apo-CIII_2_ [49]. Hiukka et al. reported an essential role of sialylation in the proinflammatory effect of LDL-bound apo-CIII on human aortic endothelial cells (HEACs). They found an increased immune response of HAECs after incubation with LDL containing apo-CIII_2_, while the immune response after incubation with LDL containing apo-CIII_0_ or apo-CIII_1_ did not differ from apo-CIII free LDL [23]. It is difficult to explain how the effects on large vessels translate to small vessel disease. Nevertheless, inflammation and dyslipidaemia are known risk factors for micro- and macrovascular complications of diabetes [50]. Moreover, higher serum sialic acid levels have been associated with inflammation [51] and coronary artery disease [52]. An increased immune response-related change in sialylation might explain our positive association of apo-CIII_2_ with retinopathy. Taken together, the association of apo-CIII sialylation with retinopathy might reflect causality. Nevertheless, the current literature does not provide a complete explanation of the pathophysiological pathway of this association.

A negative association of rs3213497-T and apo-CIII_1_ with nephropathy was found. Notably, rs3213497-T was associated with decreased apo-CIII_0a_ levels at a suggestive significance level (*p* = 1.30 × 10^−7^), but not with apo-CIII_1_ in our previous GWAS [36]. Here, only apo-CIII_1_, and not apo-CIII_0a_, presented a suggestive significance level negative association with nephropathy. The negative association of apo-CIII_1_ with nephropathy was in line with an increase in the apo-CIII_2_/Apo-CIII_1_ ratio in VLDL reported in a small sample of chronic kidney disease [53]. The associations of apo-CIII glycosylation with nephropathy did not reach significance after Bonferroni correction. Nevertheless, the directions of effect were similar to the associations of apo-CIII glycosylation with retinopathy, which is also a micro-vascular complication of diabetes, with overlapping risk factors [50].

Studied only in a hospital-treated subgroup of the DiaGene study, monosialylated apo-CIII (apo-CIII_1_) levels were positively associated with neuropathy. This trend remained in all subsequent models, independent of the duration of diabetes, glucose metabolism, the use of statins or fibrates, or insulin. Neuropathy showed an opposite direction of effect compared to retinopathy and nephropathy. Possibly, the association with apo-CIII_1_ was driven by differences in disease severity or other comorbidities of this hospital-treated subgroup.

Non-glycosylated apo-CIII (apo-CIII_0a_) was negatively associated with the prevalence of macrovascular complications. This association lost significance after the addition of lipid-reducing agents to the statistical model. The use of lipid-reducing agents was associated with lower relative levels of apo-CIII_0a_ (Appendix A). Possibly, the use of lipid-lowering medication alters the proportion of apo-CIII_0a_ and drives the association of apo-CIII_0a_ with macrovascular disease.

As for the strengths of this study, this is the first report investigating associations of apo-CIII glycosylation with micro- and macrovascular complications in type 2 diabetes. We used a high-resolution MS method to assess the apo-CIII glycosylation patterns in blood plasma within a large cohort of type 2 diabetes patients from all lines of care and replicated our genetic findings in a second cohort. The DiaGene cohort served as a high-risk population for discovery analyses, while genetic associations could even be replicated in the relatively lower-risk population of Hoorn DCS. Many clinical features of these patients were available for the analysis, allowing us to correct for possible confounders. Furthermore, we were able to investigate findings on glycosylation in the light of genetic background derived from GWASs, allowing us to assess a potential direction of effect and causality for some of the associations.

Some limitations of our study remain. The most important limitation is the amount of prospective data: the sample size for the prospective analysis was small and we had a follow-up with a median of 7.8 and 9.0 years in the DiaGene and Hoorn DCS study respectively. Due to the relatively small effect sizes, strong conclusions on effect sizes cannot be drawn. Also, the data on neuropathy were incomplete, resulting in low power for the analyses of neuropathy. The apo-CIII glycosylation profiling was performed with blood samples collected at baseline, so we were not able to assess glycosylation changes over time. Furthermore, absolute apo-CIII plasma levels were not measured; therefore, adjustment for these levels was not possible. Nevertheless, expression rates of apo-CIII_1_ and apo-CIII_2_ in humans are comparable [49], and the distribution of glycoforms in plasma is stable in variable apo-CIII concentrations in young, healthy men [54]. In coronary artery disease patients, apo-CIII_2_ remains stable across apo-CIII plasma concentrations, while apo-CIII_0c_ decreased and apo-CIII_1_ increased with increasing total apo-CIII [27]. Adjustment for total plasma apo-CIII concentrations did not affect the association of apo-CIII_2_ with plasma triglycerides, according to Koska et al. [48]. Insulin levels were not available which would have been interesting in light of the effects of apo-CIII on beta cell function. Finally, the cohort was mainly of European descent; therefore, we cannot generalise our findings to other ethnic groups.

In conclusion, our findings indicate a relationship between apo-CIII glycosylation and retinopathy, neuropathy, and macrovascular complications. In addition, a genetic variant in the *GALNT2*-gene, previously linked to increased glycosylation of apo-CIII, was found to be negatively associated with retinopathy. Future research to further investigate the possible causal pathway of retinopathy development through aberrant apo-CIII glycosylation is warranted. Together, the current study findings suggest that apo-CIII glycosylation should be considered as a potential diagnostic and therapeutic target for diabetes complications. 

## Figures and Tables

**Figure 1 ijms-25-05365-f001:**
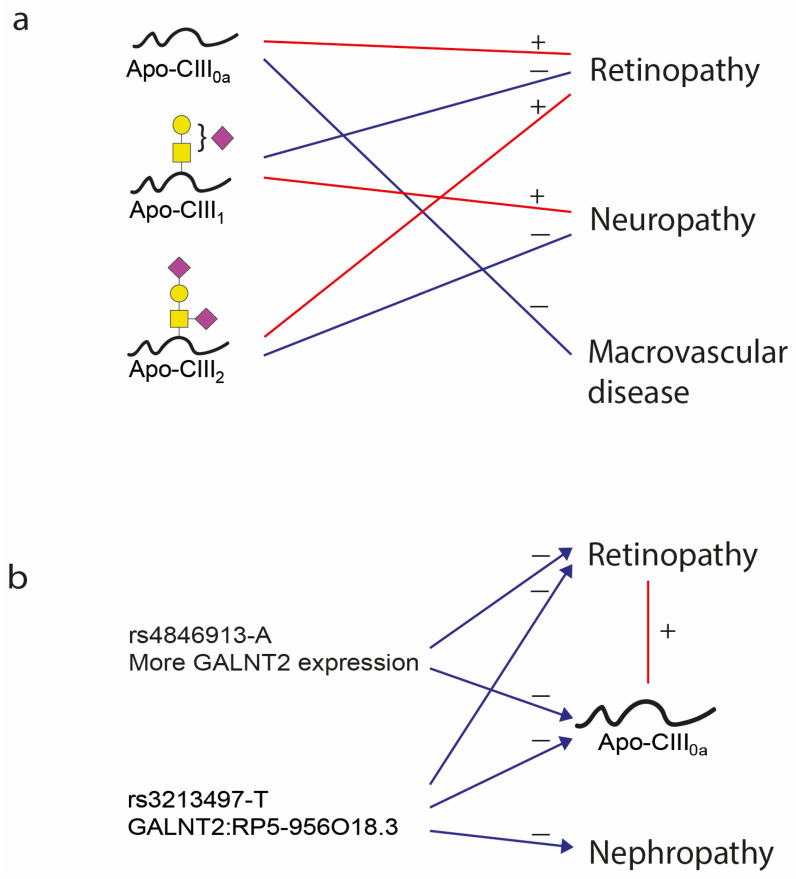
Summary of associations of genetic variants, apo-CIII proteoforms, and complications of diabetes. (**a**) Three proteoforms of apo-CIII are associated with micro- and macrovascular complications of diabetes. (**b**) Rs4846913-A is associated with less apo-CIII_0a_ and a lower prevalence of retinopathy. Rs3213497-T is associated with less apo-CIII_0a_ and decreased incidence of retinopathy and prevalence of nephropathy. Apo-CIII_0a_ is associated with a higher prevalence of retinopathy. Blue: negative associations, red: positive associations.

**Figure 2 ijms-25-05365-f002:**
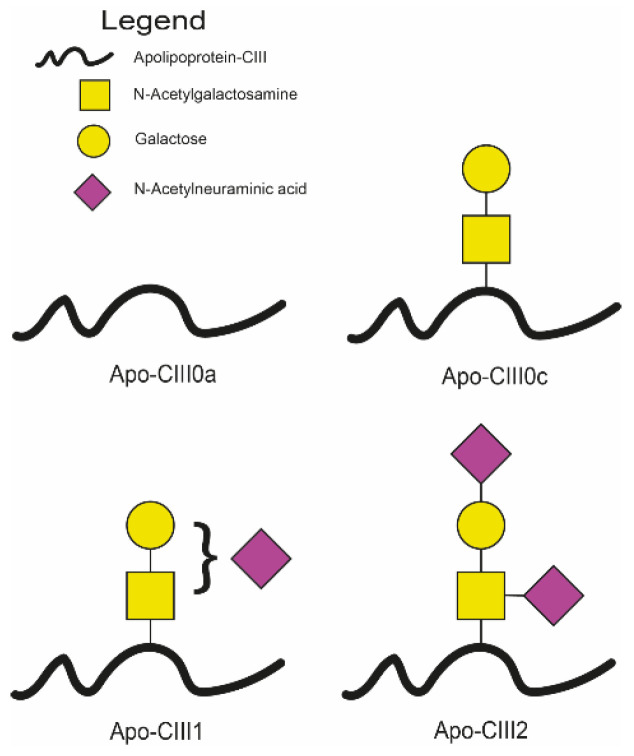
Apo-CIII glycoforms. Apo-CIII has four main proteoforms: apo-CIII_0a_ is the non-glycosylated form; apo-CIII_0c_ is glycosylated, without sialic acid; and apo-CIII_1_ and apo-CIII_2_ are glycosylated with one or two sialic acids, respectively. Black symbol = apo-CIII protein; yellow square = N-acetylgalactosamine; yellow circle = galactose; purple diamond = N-acetylneuraminic acid (sialic acid).

**Table 1 ijms-25-05365-t001:** General characteristics of the study populations.

	DiaGene (*n* = 1571)	Hoorn DCS (*n* = 5409)	Reference Values
Sex male/female (%/%)	54.2/45.8	55.4/44.6	
Age, year, mean (±SD)	65.1 (10.6)	61.1 (11)	
Duration of diabetes, year, median (IQR)	8.0 (10.6)	0.6 (2.7)	
BMI, kg/m2, median (IQR)	30.0 (6.3)	29.4 (6.5)	18.5–24.9
HbA1c, %, median (IQR)	6.8 (1.3)	6.7 (1.4)	4.0–6.0
HbA1c, mmol/mol, median (IQR)	50.8 (14.2)	49.7 (15.8)	20–42
Systolic blood pressure, mmHg, mean (±SD)	142.0 (19.0)	142.5 (20)	<140
Diastolic blood pressure, mmHg, mean (±SD)	77.5 (9.9)	80.8 (10)	<90
Mean arterial pressure, mmHg, mean (±SD)	99.0 (11.0)	101.4 (12)	<107
Creatinine, μmol/l, median (IQR)	77.0 (25.0)	79.9 (21.7)	61–120/48–99 *
HDL-cholesterol, mmol/L, median (IQR)	1.1 (0.4)	1.2 (0.4)	>1.0
Non-HDL-cholesterol, mmol/L, median (IQR)	3.0 (1.1)	3.8 (1.5)	<3.9
LDL-cholesterol, mmol/L, mean (±SD)	2.5 (0.8)	2.9 (1.5)	<3.0
Total Cholesterol, mmol/L, median (IQR)	4.2 (1.1)	5.0 (1.6)	<5.0
Triglycerides, mmol/L, median (IQR)	1.4 (1.0)	1.7 (1.1)	<2.0
*Medication use, n (%)*			
Statins or fibrates	1020 (69.2)	2276 (42.1)	
Insulin or analogues	463 (31.4)	447 (8.3)	
*Smoking, n (%)*			
Never	361 (25.4)	1667 (31.9)	
Former	804 (56.5)	2429 (46.5)	
Current	257 (18.1)	1130 (21.6)	
*Complications, n-case/n-total (%)*			
Prevalent Retinopathy	257/1478 (17.4)	194/5321 (3.6)	
Incident Retinopathy	175/1168 (15.0)	824/5127 (16.1)	
Prevalent Nephropathy	309/1401 (22.1)	499/5407 (9.2)	
Incident Nephropathy	206/1058 (19.5)	1009/4908 (20.6)	
Prevalent Neuropathy	187/611 (30.6)	NA	
Incident Neuropathy	192/400 (48.0)	NA	
Prevalent macrovascular disease	536/1328 (40.4)	111/5396 (2.1)	
Incident macrovascular disease	86/913 (9.4)	438/5285 (8.3)	

Continuous data are presented as mean (and standard deviation) or median (and interquartile range) for normal and non-normal distributions respectively. The distribution of the clinical variables was considered normal when Skewness and Kurtosis were within the range of −1 to +1. The right column shows reference values for the clinical measurements based on the guidelines for Dutch general practitioners [33,34,35]. * The reference values for creatinine are dependent on age and sex, reference values for the age group of 51–65 years are given for males and females respectively. BMI, body mass index; HDL, high-density lipoprotein; LDL, low-density lipoprotein; n, number; SD, standard deviation; IQR, interquartile range; NA, not available.

**Table 2 ijms-25-05365-t002:** Associations of apo-CIII proteoforms with prevalent complications of type 2 diabetes in the DiaGene study.

Retinopathy						
	Model 1			Model 2		
Proteoform	Beta	95% CI	*p*-Value	Beta	95% CI	*p*-Value
Apo-CIII_0a_	4.053	−3.502 to 11.609	0.293	9.968	1.437 to 18.499	**0.022**
Apo-CIII_0c_	−3.930	−9.252 to 1.392	0.148	−3.372	−9.213 to 2.468	0.258
Apo-CIII_1_	−7.215	−11.137 to −3.294	** 0.0003 **	−4.121	−8.543 to 0.301	0.068
Apo-CIII_2_	5.309	2.279 to 8.339	** 0.0006 **	3.379	−0.024 to 6.783	0.052
**Nephropathy**						
	**Model 1**			**Model 2**		
**Proteoform**	**Beta**	**95% CI**	***p*-Value**	**Beta**	**95% CI**	***p*-Value**
Apo-CIII_0a_	0.486	−6.938 to 7.910	0.898	2.403	−5.132 to 9.938	0.532
Apo-CIII_0c_	0.363	−4.519 to 5.244	0.884	0.890	−4.082 to 5.861	0.726
Apo-CIII_1_	−3.945	−7.716 to −0.174	**0.040**	−2.468	−6.338 to 1.402	0.211
Apo-CIII_2_	2.377	−0.512 to 5.266	0.107	1.386	−1.589 to 4.361	0.361
**Neuropathy**						
	**Model 1**			**Model 2**		
**Proteoform**	**Beta**	**95% CI**	***p*-Value**	**Beta**	**95% CI**	***p*-Value**
Apo-CIII_0a_	5.505	−4.277 to 15.287	0.270	5.871	−3.979 to 15.720	0.243
Apo-CIII_0c_	1.064	−5.392 to 7.520	0.747	0.951	−5.545 to 7.447	0.774
Apo-CIII_1_	7.706	2.317 to 13.095	** 0.005 **	8.000	2.558 to 13.441	** 0.004 **
Apo-CIII_2_	−4.968	−9.065 to −0.871	**0.017**	−5.116	−9.260 to −0.972	**0.016**
**Macrovascular disease**						
	**Model 1**			**Model 2**		
**Proteoform**	**Beta**	**95% CI**	***p*-Value**	**Beta**	**95% CI**	***p*-Value**
Apo-CIII_0a_	−9.195	−15.847 to −2.543	** 0.007 **	−8.795	−15.471 to −2.118	** 0.010 **
Apo-CIII_0c_	0.635	−3.615 to 4.885	0.770	0.880	−3.392 to 5.152	0.686
Apo-CIII_1_	−0.123	−3.429 to 3.184	0.942	0.301	−3.042 to 3.644	0.860
Apo-CIII_2_	−0.077	−2.622 to 2.468	0.953	−0.401	−2.979 to 2.176	0.760

Model 1: adjusted for age and sex; Model 2: adjusted for age, sex, duration of diabetes, and HbA1c. Apo-CIII_0a_ normalized to all four proteoforms: apo-CIII_0a_, apo-CIII_0c_, apo-CIII_1_, and apo-CIII_2_. The sum of the glycoforms Apo-CIII_0c_, Apo-CIII_1_, and Apo-CIII_2_ was set to 1.0. Beta represents the change of prevalence of the respective complication per increase of 1 standard deviation of the relative peak intensity of the proteoform of apo-CIII. Bold for *p* < 0.05, bold and underlined for *p* < 0.013.

**Table 3 ijms-25-05365-t003:** Associations of apo-CIII proteoforms with incident complications of type 2 diabetes in the DiaGene study.

Retinopathy						
	Model 1			Model 2		
Proteoform	Beta	95% CI	*p*-Value	Beta	95% CI	*p*-Value
Apo-CIII_0a_	−4.554	−13.457 to 4.348	0.316	3.171	−5.753 to 12.094	0.486
Apo-CIII_0c_	−5.834	−11.866 to 0.198	0.058	−4.784	−10.840 to 1.271	0.122
Apo-CIII_1_	−4.958	−9.268 to −0.649	**0.024**	−2.037	−6.521 to 2.447	0.373
Apo-CIII_2_	4.484	1.158 to 7.810	** 0.008 **	2.522	−0.962 to 6.006	0.156
**Nephropathy**						
	**Model 1**			**Model 2**		
**Proteoform**	**Beta**	**95% CI**	***p*-Value**	**Beta**	**95% CI**	***p*-Value**
Apo-CIII_0a_	−5.189	−13.555 to 3.178	0.224	−4.794	−13.196 to 3.608	0.263
Apo-CIII_0c_	−0.130	−5.465 to 5.204	0.962	−0.488	−5.831 to 4.855	0.858
Apo-CIII_1_	−2.852	−6.933 to 1.229	0.171	−2.761	−6.936 to 1.413	0.195
Apo-CIII_2_	1.366	−1.863 to 4.595	0.407	1.400	−1.877 to 4.677	0.402
**Neuropathy**						
	**Model 1**			**Model 2**		
**Proteoform**	**Beta**	**95% CI**	***p*-Value**	**Beta**	**95% CI**	***p*-Value**
Apo-CIII_0a_	−5.944	−14.462 to 2.573	0.171	−5.555	−14.206 to 3.097	0.208
Apo-CIII_0c_	3.525	−1.469 to 8.519	0.166	2.213	−2.896 to 7.323	0.396
Apo-CIII_1_	0.647	−3.527 to 4.821	0.761	0.398	−3.896 to 4.691	0.856
Apo-CIII_2_	−1.981	−5.206 to 1.245	0.229	−1.308	−4.601 to 1.984	0.436
**Macrovascular Disease**						
	**Model 1**			**Model 2**		
**Proteoform**	**Beta**	**95% CI**	***p*-Value**	**Beta**	**95% CI**	***p*-Value**
Apo-CIII_0a_	−6.948	−19.786 to 5.890	0.289	−3.852	−16.455 to 8.751	0.549
Apo-CIII_0c_	0.564	−7.660 to 8.788	0.893	0.992	−6.978 to 8.962	0.807
Apo-CIII_1_	−2.327	−8.512 to 3.857	0.461	0.244	−6.016 to 6.504	0.939
Apo-CIII_2_	0.639	−4.317 to 5.595	0.800	−1.096	−6.055 to 3.864	0.665

Model 1: adjusted for age and sex; Model 2: adjusted for age, sex, duration of diabetes, and HbA1c. Apo-CIII_0a_ normalized to all four proteoforms: apo-CIII_0a_, apo-CIII_0c_, apo-CIII_1_, and apo-CIII_2_. The sum of the glycoforms Apo-CIII_0c_, Apo-CIII_1_, and Apo-CIII_2_ was set to 1.0. Beta represents the change of incidence of the respective complication per increase of 1 standard deviation of the relative peak intensity of the proteoform of apo-CIII. Bold for *p* < 0.05, bold and underlined for *p* < 0.013.

**Table 4 ijms-25-05365-t004:** Associations of genetic variants with prevalent complications of type 2 diabetes, Model 1 (adjusted for age and sex).

Retinopathy				DiaGene			Hoorn DCS			Meta-Analysis	
Locus	rsID	EA	RA	OR	95% CI	*p*-Value	OR	95% CI	*p*-Value	OR	*p*-Value
*GALNT2*	rs4846913	A	C	0.869	0.696 to 1.086	0.218	0.935	0.763 to 1.147	0.521	0.905	0.192
*GALNT2*	rs35498929	T	C	0.884	0.637 to 1.227	0.461	0.897	0.651 to 1.236	0.508	0.891	0.324
*GALNT2:RP5-956O18.3*	rs3213497	T	C	0.991	0.716 to 1.372	0.958	0.774	0.559 to 1.071	0.122	0.875	0.283
*IFT172/NRBP1*	rs67086575	G	A	0.819	0.594 to 1.128	0.221	0.929	0.701 to 1.231	0.606	0.879	0.231
**Nephropathy**				**DiaGene**			**Hoorn DCS**			**Meta-Analysis**	
**Locus**	**rsID**	**EA**	**RA**	**OR**	**95% CI**	***p*-Value**	**OR**	**95% CI**	***p*-Value**	**OR**	***p*-Value**
*GALNT2*	rs4846913	A	C	0.912	0.988 to 0.795	0.912	1.033	0.904 to 1.179	0.637	1.020	0.730
*GALNT2*	rs35498929	T	C	0.999	0.733 to 1.360	0.993	0.903	0.734 to 1.110	0.333	0.932	0.417
*GALNT2:RP5-956O18.3*	rs3213497	T	C	0.853	0.619 to 1.175	0.331	0.819	0.667 to 1.005	0.056	0.829	**0.033**
*IFT172/NRBP1*	rs67086575	G	A	0.282	0.848 to 0.628	0.282	1.123	0.945 to 1.335	0.189	1.003	0.980
**Neuropathy**				**DiaGene**			**Hoorn DCS**			**Meta-Analysis**	
**Locus**	**rsID**	**EA**	**RA**	**OR**	**95% CI**	***p*-Value**	**OR**	**95% CI**	***p*-Value**	**OR**	***p*-Value**
*GALNT2*	rs4846913	A	C	1.270	0.948 to 1.700	0.109	NA	NA	NA	NA	NA
*GALNT2*	rs35498929	T	C	1.350	0.895 to 2.034	0.152	NA	NA	NA	NA	NA
*GALNT2:RP5-956O18.3*	rs3213497	T	C	1.161	0.765 to 1.761	0.483	NA	NA	NA	NA	NA
*IFT172/NRBP1*	rs67086575	G	A	0.989	0.664 to 1.472	0.956	NA	NA	NA	NA	NA
**Macrovascular Complications**				**DiaGene**			**Hoorn DCS**			**Meta-Analysis**	
**Locus**	**rsID**	**EA**	**RA**	**OR**	**95% CI**	***p*-Value**	**OR**	**95% CI**	***p*-Value**	**OR**	***p*-Value**
*GALNT2*	rs4846913	A	C	1.103	0.922 to 1.320	0.283	1.056	0.806 to 1.385	0.691	1.014	0.869
*GALNT2*	rs35498929	T	C	0.917	0.711 to 1.184	0.507	1.236	0.846 to 1.806	0.273	1.029	0.842
*GALNT2:RP5-956O18.3*	rs3213497	T	C	0.983	0.755 to 1.280	0.900	1.256	0.875 to 1.803	0.216	1.076	0.534
*IFT172/NRBP1*	rs67086575	G	A	0.937	0.736 to 1.194	0.601	0.840	0.572 to 1.234	0.375	0.909	0.358

Beta represents the change of prevalence of the respective complication per increase of the effect allele. Bold for *p* < 0.05. EA, effect allele; RA, reference allele; NA, not available.

**Table 5 ijms-25-05365-t005:** Associations of genetic variants with prevalent complications of type 2 diabetes, Model 2 (adjusted for age, sex, duration of diabetes, and HbA1c).

Retinopathy				DiaGene			Hoorn DCS			Meta-Analysis	
Locus	rsID	EA	RA	OR	95% CI	*p*-Value	OR	95% CI	*p*-Value	OR	*p*-Value
*GALNT2*	rs4846913	A	C	0.739	0.575 to 0.951	**0.019**	0.928	0.754 to 1.142	0.480	0.838	0.118
*GALNT2*	rs35498929	T	C	0.858	0.592 to 1.244	0.419	0.890	0.642 to 1.234	0.485	0.876	0.292
*GALNT2:RP5-956O18.3*	rs3213497	T	C	1.099	0.771 to 1.567	0.602	0.776	0.557 to 1.081	0.133	0.918	0.625
*IFT172/NRBP1*	rs67086575	G	A	0.857	0.598 to 1.226	0.398	0.934	0.703 to 1.242	0.640	0.903	0.372
**Nephropathy**				**DiaGene**			**Hoorn DCS**			**Meta-Analysis**	
**Locus**	**rsID**	**EA**	**RA**	**OR**	**95% CI**	***p*-Value**	**OR**	**95% CI**	***p*-Value**	**OR**	***p*-Value**
*GALNT2*	rs4846913	A	C	0.949	0.760 to 1.184	0.641	1.032	0.902 to 1.180	0.651	1.008	0.889
*GALNT2*	rs35498929	T	C	0.996	0.726 to 1.367	0.982	0.889	0.721 to 1.097	0.272	0.920	0.351
*GALNT2:RP5-956O18.3*	rs3213497	T	C	0.883	0.639 to 1.220	0.450	0.776	0.628 to 0.958	0.056	0.806	**0.017**
*IFT172/NRBP1*	rs67086575	G	A	0.868	0.639 to 1.189	0.366	1.118	0.938 to 1.333	0.212	1.018	0.884
**Neuropathy**				**DiaGene**			**Hoorn DCS**			**Meta-Analysis**	
**Locus**	**rsID**	**EA**	**RA**	**OR**	**95% CI**	***p*-Value**	**OR**	**95% CI**	***p*-Value**	**OR**	***p*-Value**
*GALNT2*	rs4846913	A	C	1.254	0.935 to 1.681	0.130	NA	NA	NA	NA	NA
*GALNT2*	rs35498929	T	C	1.350	0.895 to 2.038	0.153	NA	NA	NA	NA	NA
*GALNT2:RP5-956O18.3*	rs3213497	T	C	1.182	0.779 to 1.795	0.432	NA	NA	NA	NA	NA
*IFT172/NRBP1*	rs67086575	G	A	0.978	0.656 to 1.459	0.914	NA	NA	NA	NA	NA
**Macrovascular Complications**				**DiaGene**			**Hoorn DCS**			**Meta-Analysis**	
**Locus**	**rsID**	**EA**	**RA**	**OR**	**95% CI**	***p*-Value**	**OR**	**95% CI**	***p*-Value**	**OR**	***p*-Value**
*GALNT2*	rs4846913	A	C	1.091	0.911 to 1.307	0.343	1.036	0.781 to 1.374	0.806	1.075	0.354
*GALNT2*	rs35498929	T	C	0.915	0.708 to 1.182	0.495	1.255	0.844 to 1.866	0.262	1.031	0.840
*GALNT2:RP5-956O18.3*	rs3213497	T	C	0.990	0.760 to 1.289	0.939	1.216	0.831 to 1.780	0.314	1.059	0.606
*IFT172/NRBP1*	rs67086575	G	A	0.947	0.743 to 1.208	0.663	0.867	0.586 to 1.284	0.476	0.924	0.455

Beta represents the change in prevalence of the respective complication per increase of the effect allele. Bold for *p* < 0.05. EA, effect allele; RA, reference allele; NA, not available.

**Table 6 ijms-25-05365-t006:** Associations of genetic variants with incident complications of type 2 diabetes, Model 1 (adjusted for age and sex).

Retinopathy				DiaGene			Hoorn DCS			Meta-Analysis	
Locus	rsID	EA	RA	HR	95% CI	*p*-Value	HR	95% CI	*p*-Value	HR	*p*-Value
*GALNT2*	rs4846913	A	C	0.992	0.778 to 1.266	0.949	1.100	0.997 to 1.214	0.057	1.084	0.082
*GALNT2*	rs35498929	T	C	0.724	0.494 to 1.063	0.099	1.106	0.960 to 1.275	0.162	0.931	0.732
*GALNT2:RP5-956O18.3*	rs3213497	T	C	0.696	0.468 to 1.037	0.075	0.851	0.734 to 0.988	**0.034**	0.830	** 0.009 **
*IFT172/NRBP1*	rs67086575	G	A	0.916	0.654 to 1.281	0.607	1.026	0.902 to 1.168	0.693	1.011	0.855
**Nephropathy**				**DiaGene**			**Hoorn DCS**			**Meta-Analysis**	
**Locus**	**rsID**	**EA**	**RA**	**HR**	**95% CI**	***p*-Value**	**HR**	**95% CI**	***p*-Value**	**HR**	***p*-Value**
*GALNT2*	rs4846913	A	C	0.862	0.696 to 1.067	0.173	0.989	0.906 to 1.081	0.814	0.957	0.453
*GALNT2*	rs35498929	T	C	1.049	0.774 to 1.423	0.756	0.952	0.831 to 1.091	0.480	0.968	0.601
*GALNT2:RP5-956O18.3*	rs3213497	T	C	1.069	0.796 to 1.436	0.657	0.931	0.817 to 1.061	0.284	0.952	0.422
*IFT172/NRBP1*	rs67086575	G	A	0.961	0.719 to 1.286	0.791	0.987	0.876 to 1.112	0.828	0.983	0.767
**Neuropathy**				**DiaGene**			**Hoorn DCS**			**Meta-Analysis**	
**Locus**	**rsID**	**EA**	**RA**	**HR**	**95% CI**	***p*-Value**	**HR**	**95% CI**	***p*-Value**	**HR**	***p*-Value**
*GALNT2*	rs4846913	A	C	1.040	0.827 to 1.308	0.735	NA	NA	NA	NA	NA
*GALNT2*	rs35498929	T	C	1.258	0.897 to 1.766	0.184	NA	NA	NA	NA	NA
*GALNT2:RP5-956O18.3*	rs3213497	T	C	0.974	0.683 to 1.388	0.883	NA	NA	NA	NA	NA
*IFT172/NRBP1*	rs67086575	G	A	0.857	0.617 to 1.191	0.358	NA	NA	NA	NA	NA
**Macrovascular Complications**				**DiaGene**			**Hoorn DCS**			**Meta-Analysis**	
**Locus**	**rsID**	**EA**	**RA**	**HR**	**95% CI**	***p*-Value**	**HR**	**95% CI**	***p*-Value**	**HR**	***p*-Value**
*GALNT2*	rs4846913	A	C	1.020	0.731 to 1.424	0.906	1.020	0.892 to 1.166	0.773	1.020	0.751
*GALNT2*	rs35498929	T	C	0.954	0.589 to 1.548	0.850	1.128	0.930 to 1.370	0.222	1.103	0.287
*GALNT2:RP5-956O18.3*	rs3213497	T	C	1.313	0.823 to 2.096	0.253	0.885	0.722 to 1.086	0.242	1.018	0.926
*IFT172/NRBP1*	rs67086575	G	A	0.608	0.355 to 1.040	0.069	0.985	0.822 to 1.180	0.867	0.830	0.418

Beta represents the change of incidence of the respective complication per increase of the effect allele. Bold for *p* < 0.05, bold and underlined for *p* < 0.013. EA, effect allele; RA, reference allele; NA, not available.

**Table 7 ijms-25-05365-t007:** Associations of genetic variants with incident complications of type 2 diabetes, Model 2 (adjusted for age, sex, duration of diabetes, and HbA1c).

Retinopathy				DiaGene			Hoorn DCS			Meta-Analysis	
Locus	rsID	EA	RA	HR	95% CI	*p*-Value	HR	95% CI	*p*-Value	HR	*p*-Value
*GALNT2*	rs4846913	A	C	0.884	0.684 to 1.142	0.344	1.081	0.978 to 1.194	0.128	1.013	0.892
*GALNT2*	rs35498929	T	C	0.773	0.522 to 1.145	0.199	1.110	0.963 to 1.280	0.151	0.972	0.870
*GALNT2:RP5-956O18.3*	rs3213497	T	C	0.741	0.496 to 1.108	0.145	0.848	0.730 to 0.985	**0.031**	0.834	** 0.012 **
*IFT172/NRBP1*	rs67086575	G	A	0.852	0.612 to 1.187	0.344	1.033	0.907 to 1.177	0.624	0.999	0.991
**Nephropathy**				**DiaGene**			**Hoorn DCS**			**Meta-Analysis**	
**Locus**	**rsID**	**EA**	**RA**	**HR**	**95% CI**	***p*-Value**	**HR**	**95% CI**	***p*-Value**	**HR**	***p*-Value**
*GALNT2*	rs4846913	A	C	0.868	0.701 to 1.076	0.197	0.980	0.896 to 1.072	0.665	0.961	0.373
*GALNT2*	rs35498929	T	C	1.061	0.783 to 1.437	0.704	0.967	0.844 to 1.108	0.625	0.982	0.768
*GALNT2:RP5-956O18.3*	rs3213497	T	C	1.082	0.805 to 1.454	0.601	0.934	0.818 to 1.066	0.310	0.957	0.477
*IFT172/NRBP1*	rs67086575	G	A	0.968	0.725 to 1.293	0.827	0.981	0.869 to 1.108	0.760	0.979	0.714
**Neuropathy**				**DiaGene**			**Hoorn DCS**			**Meta-Analysis**	
**Locus**	**rsID**	**EA**	**RA**	**HR**	**95% CI**	***p*-Value**	**HR**	**95% CI**	***p*-Value**	**HR**	***p*-Value**
*GALNT2*	rs4846913	A	C	1.034	0.822 to 1.300	0.778	NA	NA	NA	NA	NA
*GALNT2*	rs35498929	T	C	1.285	0.917 to 1.801	0.145	NA	NA	NA	NA	NA
*GALNT2:RP5-956O18.3*	rs3213497	T	C	0.990	0.694 to 1.411	0.956	NA	NA	NA	NA	NA
*IFT172/NRBP1*	rs67086575	G	A	0.860	0.618 to 1.197	0.370	NA	NA	NA	NA	NA
**Macrovascular Complications**				**DiaGene**			**Hoorn DCS**			**Meta-Analysis**	
**Locus**	**rsID**	**EA**	**RA**	**HR**	**95% CI**	***p*-Value**	**HR**	**95% CI**	***p*-Value**	**HR**	***p*-Value**
*GALNT2*	rs4846913	A	C	0.938	0.665 to 1.323	0.716	0.999	0.872 to 1.143	0.983	0.991	0.883
*GALNT2*	rs35498929	T	C	0.906	0.551 to 1.490	0.698	1.151	0.947 to 1.399	0.156	1.116	0.236
*GALNT2:RP5-956O18.3*	rs3213497	T	C	1.387	0.865 to 2.226	0.175	0.870	0.707 to 1.071	0.189	1.045	0.848
*IFT172/NRBP1*	rs67086575	G	A	0.687	0.400 to 1.179	0.173	1.002	0.836 to 1.202	0.979	0.907	0.558

Beta represents the change of incidence of the respective complication per increase of the effect allele. Bold for *p* < 0.05, bold and underlined for *p* < 0.013. EA, effect allele; RA, reference allele; NA, not available.

## Data Availability

The data underlying this article will be shared on reasonable request to the corresponding author. The data are not publicly available due to privacy regulations.

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
