# Peer review of "Apolipoprotein-CIII O-Glycosylation Is Associated with Micro- and Macrovascular Complications of Type 2 Diabetes"

_ijms, 2024, doi:10.3390/ijms25105365_

Round 1

Reviewer 1 Report

Comments and Suggestions for Authors

The goal of this study is to investigate the relation of apo-CIII glycosylation with the prevalence and incidence of diabetic retinopathy, nephropathy, neuropathy, and macrovascular complications. They found that apo-CIII glycosylation (specifically, sialylation) and the linked GALNT2-gene variant were associated with the prevalence and incidence of diabetic retinopathy.

This project also contributes to the importance of apo-CIII glycosylation in the pathophysiology of Type-2 diabetes.

1. They specified the method that was used to determine the apo_CIII proteoforms (MALDI FT-ICR MS)in the DiaGene Study, but they did not determine the method in the Hoorn DCs.

2. If Apo-CIII glycosylation data was only available in the DiaGene study, why do they include the Hoorn Study?

2. Why did they include only females in the study?

3. Table 1 should include the normal range of each parameter so the reader can quickly assess the severity of each parameter in the Type 2 diabetic patient population.

4. Figure 2 could be presented in a less complicated, more understandable way. Maybe a table would be better?

Reviewer 2 Report

Comments and Suggestions for Authors

Paper comments

The manuscript titled “Apolipoprotein-CIII O-glycosylation is associated with micro- and macrovascular complications of type 2 diabetes, by Naber et al., details theoretical and experimental investigations.

1.      Introduction about the prevalence of type-2 diabetes and Apolipoprotein-CIII levels linked to dyslipidaemia was well written and supportive for the results and discussion.

2.      The Hoorn DCS study include 13,955 people with T2D in 2017, with follow-up ranging from 1 to 19 years. Why did the authors only consider n=5409 participants?

3.      There is a typo in page 5, line 1, word “using” repeated twice.

4.      Authors mentioned the Complications for Retinopathy was higher in the DiaGene than in the Hoorn DCS, what is the reason?

5.      Why Duration of diabetes, year, median so low in HoornDCS 0.6 years, meanwhile in DiaGene 8 years?

6.      Did you find any correlation for Apo-CIIIoc for retinopathy and neuropathy? Is there a significance for this Apo-CIIIoc?

7.      Measuring insulin level on these patients could have been more supportive for this study.

Comments on the Quality of English Language

Some minor typos and alignments needed.
